# Exercise Prescription and Progression Practices among US Cardiac Rehabilitation Clinics

**Joesi Krieger, Nicholas McCann, Markaela Bluhm and Micah Zuhl ***

School of Health Sciences, Central Michigan University, Mount Pleasant, MI 48859, USA;
jkrieger@lindenwood.edu (J.K.); mccan1nk@cmich.edu (N.M.); witcz1ml@cmich.edu (M.B.)
* Correspondence: zuhl1m@cmich.edu; Tel.: +1-989-774-7762

**Abstract:** Background: Little is known about exercise prescription practices in cardiac rehabilitation (CR). Therefore, the purpose of this study was to understand how initial exercise is prescribed and how exercise intensity is progressed among cardiac patients enrolled in United States CR programs. Methods: A 22-question survey was sent out to US CR clinics. Results: Ninety-three clinics responded to the survey. RPE was the most commonly reported exercise intensity indicator used for prescribing exercise, followed by resting HR + 20–30 bpm. Exercise progression practices were also based on patient sustained RPE values. Conclusions. Exercise prescription practice has become reliant on subjective indicators of exercise intensity. This may limit patient outcomes, such as improvement in functional measures.

**Keywords:** cardiac rehabilitation; exercise prescription; cardiovascular disease; rating of perceived exertion

## 1. Introduction

Cardiovascular disease (CVD) remains the leading cause of mortality in the United States and worldwide. Incidence is growing due to an increased prevalence of multiple risk factors, such as obesity, dyslipidemia, high blood pressure, physical inactivity, and type 2 diabetes [1]. The American Heart Association (AHA) ranked CVD as the costliest chronic disease in the US and estimated medical costs to grow from $555 billion in 2016 to $1.1 trillion by 2035 [2]. Cardiac rehabilitation (CR) is the recommended secondary prevention model to reduce all-cause mortality, hospital readmissions, and improve health-related quality of life for patients with CVD [3–5]. CR is a comprehensive intervention that integrates supervised aerobic exercise training, risk factor management (lipids, blood pressure, weight, diabetes mellitus, and smoking), nutrition education, and psychosocial support.

The primary exercise outcome of CR is an improvement in functional capacity, which is the maximal ability of an individual to perform aerobic work and defined by the product of cardiac output and arteriovenous oxygen (a-$\dot{V}o_2$) difference [6]. Functional capacity is often reported as peak metabolic equivalents (METs), where 1 metabolic equivalent represents rest and equates to 3.5 mL·kg$^{-1}$·min$^{-1}$. An increase in functional capacity of 1 mL·kg$^{-1}$·min$^{-1}$ is associated with an approximately 10% reduction of cardiac mortality [7,8]. Functional capacity is also an important indicator of prognosis (e.g., 5-year mortality) among CVD patients [9]. Specifically, each 1 MET increase in functional capacity confers a 19% reduction in CVD mortality among heart disease patients [10]. The implementation of safe and progressive exercise prescriptions during enrollment in cardiac rehabilitation is critical to the successful improvement of functional capacity.

Prominent organizations, such as the American Association of Cardiovascular and Pulmonary Rehabilitation (AACVPR), the American College of Sports Medicine (ACSM), the European Society of Cardiology, along with agencies in Asia and Australia, have developed evidence-based exercise prescription guidelines and recommendations for CVD patients enrolled in CR [11–13]. These guidelines have evolved, with the first guidelines

published in 1975 and updated eleven times [12]. Exercise intensity benchmarks include 40–80% (moderate to vigorous intensity) of heart rate reserve (HRR) or oxygen consumption reserve (VO$_2$R) in addition to subjective intensity, using rating of perceived exertion (RPE) of light to hard (12–16 on a 20-point scale). Duration goals are 20–60 min of aerobic exercise activity at a frequency of 3–5 days per week [14]. Clinicians are recommended to titrate exercise intensity with the goal of achieving the suggested target range of 40–80%; however, guidance on initial exercise prescription for patients is less clear. For patients who undergo entry graded exercise testing (GXT), achieving 40–80% intensity is much easier to define and target during initial exercise sessions. Current guidelines encourage an entry GXT because it provides an accurate baseline to establish the exercise prescription and allows clinicians to capture functional capacity outcomes, yet previous reports have shown that only 30% of CR clinics perform intake testing. In the absence of a GXT, initial exercise prescription is more challenging. Using 20–30 beats per minute (bpm) above resting heart rate (resting HR +20–30) as a starting point has been suggested, along with RPE targets of 11–14 ("light" to "somewhat hard" on a 20-point scale) [15]. It is important to understand how clinicians are prescribing exercise for patients early on in the rehabilitation process. It is also important to gain a better understanding of how exercise is progressed across 36 sessions of CR. Recommendations are vague and it is suggested to increase both duration and intensity until targeted goals are achieved [14].

The degree to which guidelines and recommendations are followed by CR programs across the United States has yet to be fully evaluated, particularly initial exercise prescription and patient progression. Recently, through a questionnaire assessment, CR clinics in Midwest US states demonstrated extensive variability in initial exercise prescription methods. Baseline stress tests prior to exercise in CR were performed in 33% of the programs. Rating of perceived exertion (RPE) was the most commonly used intensity indicator followed by heart rate reserve (HRR) and METs [15]. Overall, data is lacking on adherence and commonality of exercise prescription and progression guidelines in CR. The purpose of this study was to further identify current practices for prescribing initial exercise and techniques for progressing patients enrolled in CR clinics across the United States. This evaluation is important because it highlights opportunities to improve patient outcomes (e.g., functional capacity changes) and provides insight into improving clinical training for exercise physiologists.

## 2. Methods

Data were collected through an online survey produced using Qualtrics Survey Software (Provo, Utah). The exploratory qualitative survey was adapted from similar research studies conducted among Midwest CR programs [15] and Dutch CR programs [16,17]. The 22 questions included clinic characteristics, exercise prescription practices, and exercise progression techniques.

All outpatient CR sites in the United States were eligible for participation and locations were identified using search engines for each individual state. Four regions were created based on survey delivery and responses from each state: Northeast, Midwest, West, and South. The clinics were contacted via phone and the email address of a clinical supervisor or an individual responsible for writing exercise prescriptions was requested. The survey was then emailed to the address provided. Informed consent was explained on the first page and implied when the survey was started. The study was approved by the Central Michigan University Institutional Review Board (protocol 2019-987).

*Statistical Analysis*

Descriptive statistics (frequency, counts, percentages, means, and ranges) were used to analyze the survey data and were completed using Microsoft Excel and the built-in Qualtrics data "Reports" function. Clinic sizes were characterized as small, medium, and large, defined as <100 patients, 100–200 patients, and >200 patients treated weekly, respectively.

## 3. Results

A total of 123 surveys were sent with 93 responders for a response rate of 76%. Combined percentages in some analyses exceeded 100% if CR clinics had several answers to questions. In addition, some responders did not complete all the questions in the survey. We have identified total responders for survey questions where totals were less than 93. Seventy-three clinics completed the entire survey, and it was considered to only include completers; however, the lack of response on various questions did not impact the quality of the interpretation.

### 3.1. Clinical Characteristics

Each of the four United States regions, Northeast, Midwest, West, and South, were represented by 15% (*n* = 15), 32% (*n* = 29), 20% (*n* = 18), and 33% (*n* = 31) of respondents, respectively. Forty-four percent (*n* = 41) of the clinics were in a rural setting, with the remainder in urban (29%, *n* = 27) or suburban settings (27%, *n* = 25). All CR programs admitted patients who had a coronary artery bypass graft (CABG) procedure, a heart valve repair/replacement, a percutaneous coronary intervention (PCI), and/or a myocardial infarction (MI). Patients with heart failure (HF) were treated by most clinics (96%, *n* = 89). Seventy percent (*n* = 65) of programs reportedly admitted patients with a heart or lung transplant, including those with left ventricular assist devices. Sixty-seven percent (*n* = 62) reported treating patients with peripheral artery disease (PAD).

Table 1 lists CR program characteristics. Eighty-seven responders provided details about the size of their programs, with 64% classified as small, 24% as medium, and 11% as large, treating more than 200 patients per week. When asked about the individuals writing exercise prescriptions, most clinics responded by reporting numerous types of clinicians, including, exercise physiologists (84%), nurses (40%), physicians (including cardiologists) (18%), and "other" (6%). The "other" category included respiratory therapists and those described as exercise specialists. Furthermore, 78% of CR clinics (87 total responders) required professional certification (i.e., ACSM, AACVPR). Ninety-one clinics answered questions about the use of baseline testing. Of these, 50% (*n* = 45) of the clinics reported use of submaximal testing, including the 6MWT, while only 2% (*n* = 2) performed baseline graded exercise tests (GXTs). Twenty-four percent (*n* = 22) of clinics had patients complete peak MET estimation questionnaires to determine functional capacity. The remainder did not report a functional capacity assessment (24%, *n* = 22).

**Table 1.** Cardiac rehabilitation program characteristics.

| Characteristic | Results |
| --- | --- |
| Clinic size (87 responders) | Small (*n* = 56), medium (*n* = 21), large (*n* = 10) |
| Writing exercise prescriptions | CEP (*n* = 74), nurses (*n* = 35), physicians (*n* = 9), cardiologists (*n* = 7), other (*n* = 6) |
| Certification requirements (89 responders) | Yes (*n* = 69), no (*n* = 20) |
| Type of baseline testing (91 responders) | Submaximal (*n* = 45), GXT (*n* = 2), Estimation Questionnaire (*n* = 22), none (*n* = 22) |

CEP = clinical exercise physiologist; Clinic size: small (<100 patients/week), medium (100–200 patients/week), large (>100 patients/week); GXT = graded exercise test.

### 3.2. Prescription Practices

Information on initial exercise intensity prescription practices is presented in Table 2. Seventy-eight percent of CR programs used more than one variable to monitor intensity. Borg's rating of perceived exertion (RPE) was the most reported variable (93%) used for exercise prescription. Several clinics provided a Borg 0–10 range instead of a 6–20 Borg scale range. To maintain consistency, 0–10 ranges were converted to the 6–20 Borg scale so that a range of 3–7 (CR10) was equivalent to 11–15 (Borg 6–20). A rating of "fairly light"/"moderate" to "hard" (RPE of 11–15) was targeted by 62 clinics that used RPE. Fifty-

three clinics used a heart rate measure to prescribe initial exercise intensity. Resting HR +20–30 bpm was the most reported (*n* = 31), followed by 50–85%HRR (*n* = 22). A targeted initial workload was reportedly used by 18 clinics and included 2–4 METs and watts (if using a cycle ergometer). Only two clinics used %VO$_2$R, and another 10 clinics (13%) reported other methods. Other methods included use of blood pressure measurement or rating of dyspnea. Seventy-three clinics responded to questions about exercise modality with 89% (*n* = 65) using multiple modes on the first day and 11% using one mode of exercise (not reported in the table). Nearly all clinics began patients on a treadmill, followed by recumbent cycling and cycle ergometer.

**Table 2.** Initial exercise intensity and exercise progression practices.

| Characteristic | Results |
| --- | --- |
| Initial target intensity | RPE: 11–15 (*n* = 62) *, resting HR +20–30 (*n* = 31), %HRR: 50–85% (*n* = 22), workload (*n* = 18) **, 40–80% %VO$_2$R (*n* = 2), other (*n* = 7) *** |
| Indicator for exercise progression (ranked order) | Sustained RPE (*n* = 53), sustained hemodynamic response (*n* = 29), predetermined number of sessions (*n* = 11) |
| Progression order (77 responders) | Duration first (*n* = 60), intensity first (*n* = 10), other (*n* = 7) |
| Frequency of progression documentation (73 responders) | Per session (*n* = 59), weekly (*n* = 13), monthly (*n* = 3), as it occurs (*n* = 1) |
| Progress of telemetry (76 responders) | Yes (*n* = 27), no (*n* = 49) |

RPE = rating of perceived exertion; HR = heart rate; %HRR = percent heart rate reserve (Karvonen); APMHR = age predicted max heart rate; %VO$_2$R = percent maximal volume of oxygen consumption reserve. * Eighteen clinics reported RPE values using the Borg CR10. To be consistent, the values were converted to the Borg 6–20 scale, so that a rating of 3–7 (CR10) is equivalent to 11–15 (Borg 6–20). ** Workload methods included a target of 2–4 METs and watts. *** Other methods included blood pressure and dyspnea level.

### 3.3. Progression Practices

Exercise progression was individualized in nearly all programs and general progression practices are presented in Table 2. Patients were largely progressed when they reported low, sustained RPE values less than "moderate" intensity (56%, *n* = 53). Progression also occurred when hemodynamic values were sustained (32%, *n* = 29) and when a certain number of exercise sessions were completed (12%, *n* = 11).

Seventy-seven clinics responded to questions about exercise progression order. Duration was the first variable to be progressed before intensity in 78% (*n* = 60) of clinics, followed by intensity first (13%, *n* = 10). Most clinics documented intensity variables every session (81%, *n* = 59 out of 73 responders) followed by weekly (18%, *n* = 13) and monthly documentation (4%, *n* = 3). Patients were progressed off of telemetry in 36% (*n* = 27 out of 76 responders) of clinics.

### 3.4. Clinic Size and Influence of Location

Exercise prescription and patient monitoring did not vary substantially based on clinic size and location. Of note, large clinics (defined as >200 patients per week) were reportedly less likely (10%) to progress patients off of telemetry. Encouragingly, the size of the clinic did not influence progression practices, with small, medium, and large clinics reporting 75%, 85%, and 80% progression per session, respectively. Similarly, progression practices did not differ based on the location of the clinics (Northeast, Midwest, West, and South).

### 4. Discussion

The results of the survey demonstrate the variation that exists among United States CR programs with respect to exercise prescription and progression. While prescription guidelines and recommendations have been previously evaluated in Midwest states, initial exercise programming and progression of exercise in cardiac rehabilitation has been studied to a lesser extent. The results of the current study support the previous finding of variability

among exercise prescription methodology and a skewing towards RPE use as an intensity measure [15]. Similarly, patient progression techniques varied across surveyed clinics with a high reliance on RPE for making exercise adjustments. The popularity of RPE in CR is concerning for several reasons. Establishing safe and effective exercise programming should be rooted in physiological responses to exercise and not centered on subjective indicators. Further, the reliance on RPE may limit patient outcomes (e.g., improvement in functional capacity).

*4.1. Clinical Characteristics*

Traditional phase II center-based CR is commonly delivered in 1 h aerobic exercise sessions, 2–3 days a week, for a total of 36 sessions. It is approved for CABG, STEMI and NSTEMI in the past 12 months, PCI, chronic stable angina, heart valve repair or replacement, HF with reduced ejection fraction, heart or heart/lung transplant, and PAD. All clinics treated some combination of these patient groups, with PAD being the least reported group. The approach to aerobic exercise for PAD is markedly different in comparison to all other groups and demonstrates the requirement of understanding complexity and overlap of each CVD diagnosis when writing the exercise prescription and progressing patients. Despite exercise considerations among PAD patients, 67% of surveyed clinics were accepting and treating PAD patients. This is in contrast with earlier surveys among US Midwest CR programs and Dutch programs in which clinics reported a lack of PAD enrollment [15,16]. This is a promising change and possibly indicates that CR referral for PAD patients is improving.

It is also important to highlight the enrollment of patients with heart–lung transplant, and specifically those patients with ventricular assist devices (VAD). Accumulating data suggests that exercise is safe and beneficial for VAD patients, and in response new exercise recommendations have been documented [18–20].

Cardiac rehabilitation classes require physician oversight; however, physical presence is not necessary. Thus, sessions are led by other health professionals, including clinical exercise physiologists, nurses, exercise specialists, and respiratory therapists. Exercise physiologists and nurses may become certified through the same professional organizations that write the guidelines for CR (i.e., ACSM, AACVPR). In the current study, approximately 78% of surveyed clinics required professional certification. Comparatively, in an earlier analysis of Midwest States only 26% of surveyed clinics required certification [15]. This is an important shift in the field and indicates an emphasis being placed on certification standards.

Baseline and post-rehab functional capacity testing, either maximal or submaximal, is a valuable tool in the determination of patient starting points, evaluation of exercise hemodynamics, as well as monitoring improvements gained from cardiac rehabilitation [14]. Peak or maximal graded exercise testing (GXTs) is the gold standard for the determination of patient functional capacity and allows for the best determination of direct physiological intensity variables, such as HR values [10]. Abnormal physiological responses to exercise (i.e., ischemia, hypertensive blood pressure) may also be uncovered and limitations may accordingly be placed on patients' hemodynamic values during exercise. Half of the clinics reported completion of submaximal baseline testing while only two reported using GXTs at patient intake. Submaximal testing does offer some advantages, such as improved safety and ease of integration into practice, along with allowing clinicians to examine a patient's hemodynamic responses to exercise before and after completion of CR [21]. However, submaximal testing limits the ability of the clinician to target precise exercise intensity for prescription purposes. Limitations to baseline GXTs include cost, lack of equipment and trained individuals to conduct test, and patient hesitancy, which may explain the lack of testing [21]. Ideally, the safety and effectiveness of initial prescription and subsequent progression would be evaluated using a test–retest method. However, due to the listed limitations, this is not always possible, and clinicians should be trained to adequately prescribe exercise using various techniques in the absence of a baseline stress test [22]. Of concern is the high rate of reported use of precent heart rate reserve (%HRR) for prescribing

exercise intensity in the absence of a GXT. Not adequately measuring a patient's peak heart rate makes the heart rate reserve calculation obsolete for cardiac patients. Among healthy populations, the use of maximal heart rate estimation equations (e.g., 220-age) is common for prescribing %HRR and percent heart rate max (%HR$_{max}$) targets; however, estimation equations should not be used among CVD patients [22].

*4.2. Prescription Practices*

Aerobic exercise is a core component of CR as it improves cardiorespiratory fitness, disease-related symptoms, and coronary risk factors [23]. Relative intensity during aerobic exercise in CR is determined using multiple strategies: %HR$_{max}$, %HRR, METs, %VO$_2$R, and RPE. Most surveyed clinics used more than one of these strategies, with the most frequently used marker being RPE. Over 90% of programs used RPE to determine initial exercise prescriptions. The standardized subjective measure of effort, Borg's RPE, has been validated as an effective tool that is strongly correlated with several physiological variables, including HR. When used in a clinical population, however, the HR and RPE correlation may be skewed towards a higher RPE with a lower HR due to debilitations and medications [24]. The common use of this variable may be due to its ease and simplicity of use compared to other methods requiring a baseline test. For RPE use to be reliable, a patient must understand the explanation of RPE correctly. Consistent and accurate descriptions of RPE by health care professionals are necessary for universal use and understanding. Educational initiatives and standard quality control should be implemented to guarantee that all health professionals in CR can clearly explain the RPE scale. One thought is to incorporate staged RPE testing that allows the patient to experience exercise at various RPE values. In addition, RPE cannot be used solely to set intensity benchmarks and should be combined with heart rate, exercise workload, and patient symptoms. Additional input from other variables will help determine if reported RPE values are accurate. One subset of patients in CR requiring additional consideration includes those with intellectual disabilities or diminished cognitive functioning. While research is lacking, most individuals with intellectual disabilities in a previous study were able to effectively rate perceived exertion, however, others gave the same value as workload increased or provided inconsistent values [25].

As previously discussed, exercise tests are not always readily available so the use of other methods, such as RPE, is required. In the absence of a stress test, it is recommended to use either resting HR + 20–30 bpm or an RPE of 12–16 on Borg's RPE scale for outpatient CR. Generally, an RPE of 12–16 is consistent with 40–80% exercise capacity determined using exercise testing and therefore can be a valuable intensity measure in the absence of a GXT when used and understood correctly [14,26]. Of the surveyed clinics, the majority (87%) targeted an RPE of 11–15, thus many fell within the recommended range. Thirty-one clinics were using resting HR + 20–30 bpm; however, there is evidence to suggest that exercising at this intensity does not provide adequate stimulus. Reed et al. reported that exercising at resting HR + 20–30 bpm resulted in an exercise intensity of 40–60% HRR in only 26% of cardiac patients [27].

*4.3. Progression Practices*

Functional capacity improvements made during cardiac rehabilitation can improve health outcomes. Feuerstadt et al. observed that with every 1 MET increase during CR, mortality was reduced by approximately 28% [28]. Exercise training below 3.5 METs after 4 weeks of CR was associated with 7% and 18% higher 1- and 3-year mortality rates, respectively [29]. Importantly, METs at the end of 36 visits of CR is a stronger predictor of all-cause mortality than METs at program start [21]. These findings indicate the importance for health professionals in CR to continually evaluate patients for progression readiness and implement strategies that optimize MET progression throughout all 36 sessions. While not all patients will realistically reach 3.5 METs by 4 weeks into CR, even a 1 MET increase by CR completion is associated with improved outcomes.

Nearly all responding clinics individualize progression which is expected due to the considerable differences between disease categorizations and individuals. When asked how often exercise is progressed, answers provided demonstrated extreme variability; therefore, there appears to be a lack of clear and consistent timing benchmarks that maximize progression opportunities. It is necessary to closely monitor the progress of each patient to determine each opportunity for progression of workload. Of the surveyed clinics, 81% documented progression, including documentation if there was a lack of progression each session, and another 18% did so weekly. Indicators used to initiate exercise progression also varied, with sustained RPE as the most common, followed by sustained hemodynamics. Again, the reliance on RPE to progress patient workload is concerning. When used for determining whether patients are ready for progression, one matter of concern is patient complacency, where the same RPE value is given out of habit instead of actual perceived exertion. In addition, a patient may report variable RPE values despite similar workloads for several consecutive exercise sessions. This may lead to lack of adequate progression and compromise patient outcomes. Clinical guidelines do suggest the use of subjective indicators of intensity, but these should be used as adjunctive measures to help guide exercise [30].

According to ACSM, any one or a combination of the FITT principle components—frequency, intensity, duration (time), and type—may be used to progress patients through an exercise program [14]. Duration was progressed first (78%), before intensity (13%), in responding CR clinics. First progressing duration over intensity, particularly in sedentary and frail individuals, may be more beneficial for long-term outcomes. Hansen et al. found that increasing intensity during the early stages of CR in previously sedentary individuals resulted in lower adherence compared to progressing duration first [31].

Electrocardiographic monitoring with cardiac telemetry during aerobic exercise in phase II CR is common practice; however, recommendations vary in terms of number of sessions where monitoring is necessary. The majority (64%) of the responding clinics reported that patients were kept on telemetry for all 36 sessions. Ultimately, judgement of telemetry monitoring should remain with the clinical supervisor of cardiac rehabilitation patients. Evaluations should be carried out on a case-by-case basis, with consideration for the patient, staff, and setting. General guidelines do suggest that patients can be progressed off of telemetry after 6–12 sessions if they have known stable CVD and low risk for complications. Those with known CVD and with moderate-to-high risk for complications can be progressed to intermittent or no monitoring following 12 sessions. In both cases, patients can be progressed off telemetry sooner if deemed appropriate by clinical staff [26]. In conclusion, individual patient circumstances require health professionals to maintain clinical freedom when making decisions about the application, modification, and replacement of guidelines.

### 4.4. Future Direction and Limitations

There is a high level of variability in prescription and progression practices despite guidelines. That having been said, it is important to understand that prescription and progression is relative to initial patient status and individual physical, emotional, and psychological responses to exercise. However, it is important to note that attempts should be made to progress all patients to higher workloads. Holding patients at the same workloads throughout rehabilitation limits cardiovascular adaptations. This also calls attention to the importance of establishing consistent outcome measures for patients who attend CR. The AACVPR has created a registry with the aim of capturing outcome measures across clinics in the United States. The registry may help to determine valuable measures and set benchmarks (e.g., peak MET changes). Another interesting approach is to balance rigid exercise prescription practices with more emphasis on patients becoming more physically active in their daily life. This may be achieved through education, access to technology, and creating alternative exercise environments (e.g., using outdoor walking paths and encouraging game play).

Limitations of the current study include incomplete responses to all survey questions by participating programs. Potential response bias, where respondents may have been more inclined to respond in a socially desirable manner, cannot be ruled out; however, programs were informed that responses were anonymous. There was also a low number of participating clinics, and therefore it is possible that the survey respondents may not reflect the field of CR as a whole. Patient characteristics (age, sex, diagnoses) could not be determined from the collected data, and therefore understanding prescription practices for particular populations is not clear.

## 5. Conclusions

The current exploratory qualitative study supports previous findings of significant variation in exercise prescription and emphasizes the variability and complexity of progressing patients. More must be known about the impact of each progression indicator on patient safety and outcomes in diverse contexts. Future studies should focus on the development of strategies to bridge the gap between knowledge and practice to improve the quality and uniformity of cardiac rehabilitation across the United States while maintaining the individualized care that is required for the diverse set of patients. Furthermore, an understanding of exercise guidelines and considerations is needed for special cardiac populations, such as VAD patients and those with rare cardiac diseases, such as cardiac amyloidosis, which commonly lead to end-stage treatments (e.g., transplant, VAD implantation) [32]. Lastly, it is important to further understand how CR guidelines are being practiced in countries outside of the United States.

**Author Contributions:** Conceptualization, J.K., N.M., M.B. and M.Z.; methodology, M.Z.; software, J.K., N.M. and M.B.; formal analysis, J.K., N.M., M.B. and M.Z.; writing—original draft preparation, J.K.; writing—review and editing, J.K. and M.Z.; supervision, M.Z.; All authors have read and agreed to the published version of the manuscript.

**Funding:** This research received no external funding.

**Institutional Review Board Statement:** The study was approved by the Central Michigan University Institutional Review Board (protocol 2019-987).

**Informed Consent Statement:** Informed consent was obtained from all subjects involved in the study.

**Data Availability Statement:** Data shared in the article are in accordance with the consent provided by participants.

**Conflicts of Interest:** The authors declare no conflict of interest.

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
