# Peer review of "Exercise Prescription and Progression Practices among US Cardiac Rehabilitation Clinics"

_clinpract, doi:10.3390/clinpract12020023_

Round 1

Reviewer 1 Report

In this paper, authors repoted a survey on exercise prescription practices in cardiac rehabilitation in United States. Some comments:

1) in the Introduction, I would suggest to add also the European Guidelines on cardiovascular Prevention signed in 2021, where the concept of cardiac rehabilitation is well stressed too, as in the American ones;

2) authros reported that seventy percent (n=65) of programs reportedly admitted patients with a heart or lung transplant, including those with left ventricular assist devices. I think that this point deserves some comments in the Discussion, as far as the idea of physical activities in patients with ventricular assist devices is quite new, see for reference: "Di Nora C. Role of Cardiac Rehabilitation After Ventricular Assist Device Implantation. Heart Fail Clin. 2021 Apr;17(2):273-278. doi: 10.1016/j.hfc.2021.01.008"

3) it would be interesting to have some more information on the patients treated with cardiac rehabilitation, especially related to age, sex, type of cardiomyopathy....if these data are not available, please add this point as Limitation;

4) authors discussed about cardiac rehabilitation in heart transplant; however a special mention deserves the importance of physical program activities for all those patients heart transplanted for systemic disease, see: "Livi U. Heart transplantation in cardiac storage diseases: data on Fabry disease and cardiac amyloidosis. Curr Opin Organ Transplant. 2020 Jun;25(3):211-217", where the muscular involvement could be explained by the priamry systemic disorder, so the importance of the cardiac rehabilitation is even more important tahn in other cases.

Author Response

In this paper, authors reported a survey on exercise prescription practices in cardiac rehabilitation in United States. Some comments:

Response: Thank you for your thorough review of our manuscript.  We have attempted to respond to each of your comments/suggestions. 

1)in the Introduction, I would suggest to add also the European Guidelines on cardiovascular Prevention signed in 2021, where the concept of cardiac rehabilitation is well stressed too, as in the American ones;

Response: Thank you for this comment.  We have added the European guidelines and reference (lines 44-47).

2) authors reported that seventy percent (n=65) of programs reportedly admitted patients with a heart or lung transplant, including those with left ventricular assist devices. I think that this point deserves some comments in the Discussion, as far as the idea of physical activities in patients with ventricular assist devices is quite new, see for reference: "Di Nora C. Role of Cardiac Rehabilitation After Ventricular Assist Device Implantation. Heart Fail Clin. 2021 Apr;17(2):273-278. doi: 10.1016/j.hfc.2021.01.008"

Response: Great point.  We have added several statements to the discussion and included several references (lines 208-211).

3) it would be interesting to have some more information on the patients treated with cardiac rehabilitation, especially related to age, sex, type of cardiomyopathy....if these data are not available, please add this point as Limitation;

Response: These data are not available, and therefore, a statement has been added to the limitations section (lines 350-352).

4) authors discussed about cardiac rehabilitation in heart transplant; however a special mention deserves the importance of physical program activities for all those patients heart transplanted for systemic disease, see: "Livi U. Heart transplantation in cardiac storage diseases: data on Fabry disease and cardiac amyloidosis. Curr Opin Organ Transplant. 2020 Jun;25(3):211-217", where the muscular involvement could be explained by the primary systemic disorder, so the importance of the cardiac rehabilitation is even more important tahn in other cases.

Response: Thank you for the suggestion.  We have added a statement in the conclusion section about rare cardiac diseases and considerations for these populations in CR, along with appropriate references (lines 360-364).

Reviewer 2 Report

In the present study, Dr. Krieger and colleagues evaluated exercise prescription and progression practices among the cardiac rehabilitation clinics in the United States through a 22-question survey.

The authors have found significant variation in exercise prescription and have clarified the urgent need of objective indicators of exercise intensity and outcomes.

The current manuscript is interesting. This Reviewer has just a couple of comments:

-          It may be useful to discuss and reference cardiac rehabilitation experience outside the United States.

-          The authors should emphasize the differences between centers or regions (Northeast, Midwest, West, and South) in terms of results.

Author Response

In the present study, Dr. Krieger and colleagues evaluated exercise prescription and progression practices among the cardiac rehabilitation clinics in the United States through a 22-question survey.

The authors have found significant variation in exercise prescription and have clarified the urgent need of objective indicators of exercise intensity and outcomes.

Response: Thank you for your thorough review and recommendations.  We have attempted to adequately address each of your comments.

The current manuscript is interesting. This Reviewer has just a couple of comments:

-          It may be useful to discuss and reference cardiac rehabilitation experience outside the United States.

Response:  Thank you for the comment.  We have highlighted the new guidelines brought by European Cardiology Society along with Asia and Australia (with respected references) (lines 44-46).  We also added a statement about the importance of further understanding CR practices in countries outside of the United States (364-365).

-          The authors should emphasize the differences between centers or regions (Northeast, Midwest, West, and South) in terms of results.

Response: Thank you for the comment.  We did analyze data based on the location of clinics and did not detect any differences among clinics located in the Midwest, South, Northeast, and West.  A brief statement has been included in the revised manuscript (lines 178-180).

Round 2

Reviewer 1 Report

The paper has been improved after the last revisions.

Reviewer 2 Report

In the present study, Dr. Krieger and colleagues evaluated exercise prescription and progression practices among the cardiac rehabilitation clinics in the United States through a 22-question survey.

The authors have found significant variation in exercise prescription and have clarified the urgent need of objective indicators of exercise intensity and outcomes.

The manuscript has been improved during the revision process.